# MYH7 Genotype–Phenotype Correlation in a Cohort of Finnish Patients

**Teemu Vepsäläinen [1],\*, Tiina Heliö [2], Catalina Vasilescu [3], Laura Martelius [4], Sini Weckström [2], Juha Koskenvuo [5], Anita Hiippala [1]** 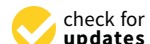 **and Tiina Ojala [1]**

1 Department of Pediatric Cardiology, Helsinki University Hospital and The University of Helsinki, 00029 Helsinki, Finland; anita.hiippala@hus.fi (A.H.); tiina.h.ojala@hus.fi (T.O.)
2 Heart and Lung Center, Helsinki University Hospital and The University of Helsinki, 00026 Helsinki, Finland; tiina.helio@hus.fi (T.H.); sini.weckstrom@hus.fi (S.W.)
3 Stem Cells and Metabolism Research Program, Biomedicum, University of Helsinki, 00290 Helsinki, Finland; catalinavasilescu@gmail.com
4 Department of Radiology, HUS Medical Imaging Center, Helsinki University Hospital and The University of Helsinki, 00029 Helsinki, Finland; laura.martelius@hus.fi
5 Blueprint Genetics, 02150 Espoo, Finland; juha.koskenvuo@blueprintgenetics.com
\* Correspondence: teemu.vepsalainen@hus.fi

**Abstract:** Cardiomyopathies (CMPs) are a heterogeneous group of diseases, frequently genetic, affecting the heart muscle. The symptoms range from asymptomatic to dyspnea, arrhythmias, syncope, and sudden cardiac death. This study is focused on MYH7 (beta-myosin heavy chain), as this gene is commonly mutated in cardiomyopathy patients. Due to the high combined prevalence of MYH7 variants and severe health outcomes, it is one of the most frequently tested genes in clinical settings. We analyzed the clinical presentation and natural history of 48 patients with MYH7-related cardiomyopathy belonging to a cohort from a tertiary center at Helsinki University Hospital, Finland. We made special reference to three age subgroups (0–1, 1–12, and >12 years). Our results characterize a clinically significant MYH7 cohort, emphasizing the high variability of the CMP phenotype depending on age. We observed a subgroup of infants (0–1 years) with MYH7 associated severe DCM phenotype. We further demonstrate that patients under the age of 12 years have a similar symptom burden compared to older patients.

**Keywords:** MYH7; cardiomyopathy; screening; age of onset



## 1. Introduction

Cardiomyopathies (CMPs) are a heterogeneous group of diseases affecting the heart muscle characterized by alterations in the ventricle wall thickness, size of the cardiac chambers, or abnormal contraction in the absence of underlying heart disease. The symptoms range from asymptomatic to dizziness, palpitations, syncope, and sudden cardiac death (SCD) [1]. CMPs can be divided into specific morphological and functional phenotypes, including dilated, hypertrophic, restrictive, arrhythmogenic, and non-compaction patterns and they frequently have a genetic basis [2,3].

This study focuses on the MYH7 gene (beta-myosin heavy chain), encoding myosin-7, which is a major contributor to several CMPs and is often associated with poor prognosis [4,5]. This large gene is located on chromosome 14 q11.2–q13 in humans (40 exons), and its product was the first sarcomeric protein linked with cardiomyopathy [6]. It is expressed predominantly in the human ventricle and skeletal muscle tissues. Mutations in MYH7 are associated with hypertrophic cardiomyopathy (HCM), dilated cardiomyopathy (DCM), left ventricular non-compaction (LVNC), restrictive cardiomyopathy (RCM), and multiple patients have shown overlapping phenotypes. In addition, MYH7 mutations can cause a combination of myopathy and cardiomyopathy, and pure skeletal myopathies.

Due to MYH7 high combined prevalence and severe health outcomes, it is one of the most frequently tested genes in a clinical setting [4,5,7].

This study analyzes the clinical presentation and natural history of patients with MYH7-related cardiomyopathy with special reference to patients under 12 years of age. For this purpose, we investigated a cohort of MYH7-related CMPs from a tertiary center in Helsinki University Hospital, Finland. We present here the clinical and genetic characteristics of the MYH7 cohort and demonstrate that patients under the age of 12 years have a similar symptom burden to older patients. Furthermore, we noticed a subgroup of infants with MYH7, which associates with a severe DCM phenotype.

## 2. Materials and Methods

Our retrospective study aimed to include all identified MYH7 variants in the Helsinki University Hospital district in Finland between 2002 and 2021. Patients under 16 were followed and diagnosed at the children's hospital and older patients at the adults' cardiology department. Five patients declined their participation in the study. The final study population consisted of 21 index patients and 27 first-degree relatives. Of these 27 first-degree relatives, nine were not related to any other study patients.

Cardiac imaging data were collected for each patient. All patients underwent echocardiography and resting and ambulatory electrocardiograph as a part of their clinical workup. Cardiac magnetic resonance imaging (CMR) was performed on selected patients to verify CMP diagnosis and to assess the severity of the disease [8]. Cardiac MRI was performed using a 1.5 Tesla clinical system (Ingenia, Philips Healthcare, Best, The Netherlands in pediatric patients and Siemens MAGNETOM Avanto/Avantofit, Siemens Healthcare in adult patients) and was in line with current ESC recommendations [9]. Imaging investigations were performed by a senior radiologist (CMR) or cardiologist (CMR, echocardiography) with a specialization in cardiac imaging settings. Cardiac echocardiography was performed using a GE Healthcare or Philips Healthcare ultrasound device. DCM was defined by reduced ventricular systolic function: fractional shortening <25%, ventricular ejection fraction (EF) <45%, and left ventricular end-diastolic diameter >27 mm/m$^2$, or in children a Z score $\geq$+2. HCM was defined by otherwise unexplained septal hypertrophy, left ventricular free wall hypertrophy $\geq$15 mm, or in first-degree relatives of HCM $\geq$13 or in children Z score $\geq$+2 [9]. For LVNC, CMR was used in all patients to confirm the diagnosis of LVNC and to examine the disease's severity. The following MRI criterion was used: the end-diastolic ratio of non-compacted to compacted layers above 2.3 in short axis view [10]. No patients were detected with MYH7 related arrhythmogenic right ventricular cardiomyopathy.

Clinical data were collected from routine outpatient visits from all the patients, including heart failure symptoms according to the New York Heart Association (NYHA) or Ross functional classification [11,12], family history, resting and ambulatory electrocardiography (ECG), and transthoracic echocardiography (2D, Doppler, and color). Heart failure symptoms were defined as an NYHA or Ross functional class $\geq$2. Left ventricular outflow tract obstruction (LVOTO) was defined as an instantaneous peak Doppler left ventricular outflow tract pressure gradient $\geq$30 mmHg at rest [9]. Non-sustained ventricular tachycardia (NSVT) was defined as three or more consecutive ventricular beats at a rate greater than 120 beats/min lasting less than 30 s on ambulatory ECG recordings [9]. The patient was considered to have familial cardiomyopathy if at least one first-degree or two second-degree relatives were reported to have cardiomyopathy. Sudden cardiac death was defined as a sudden and unexpected death. This information was obtained from the patients during a routine clinical visit as a part of family history.

Genetic testing was performed as part of usual care. Of the 48 patients, the NGS panel method was used for 34, and 12 were separately diagnosed for their gene defect. As part of usual clinical practice, targeted sequencing panels vary according to the phenotype presented by patients (e.g., HCM, DCM). The targeted panels used were the Cardiomyopathy Panel version 1.0–3.0 Panel (Blueprint Genetics, Helsinki, Finland), 155 cardiomyopathy-

related genes; Hypertrophic Cardiomyopathy (HCM) Panel (Blueprint Genetics, Helsinki, Finland), 16 genes; Pan cardiomyopathy panel (Blueprint genetics, Helsinki, Finland), 103 genes; Dilated Cardiomyopathy (DCM) Panel Plus, (Blueprint genetics, Helsinki, Finland), 69 genes. Furthermore, whole-exome sequencing was performed for two patients. All the gene findings were verified with bi-directional Sanger sequencing. When an MYH7 proband was diagnosed, first-degree relatives were tested clinically independent of age by Sanger sequencing.

All variants initially considered disease-causing were re-analyzed and classified for this study according to the American College of Medical Genetics and Genomics (ACMG) guidelines as pathogenic (P), likely pathogenic (LP), variant of unknown significance (VUS), likely benign (LB), or benign (B) [13]. Variants classified as P/LP were considered to be diagnostic. Pathogenic, likely pathogenic, and VUS variants were included in the study (Table 1). We excluded from the analyses one patient whose variant was classified as benign MYH7 c. 4472 C > G, *p.* (Ser1491Cys) and one patient who had wide homozygous regions in the exome. One patient also had another potential cardiomyopathy variant (DSG2 c. 2137 G > A, *p.* (Glu713Lys)), but the clinical phenotype did not support this variant as the patient's underlying cause of HCM, so the patient was included in the study.

**Table 1.** MYH7 variants observed in the study patients.

| Gene | Sex | CMP | Age at the Time of Dg (Yrs) | Classification | Endpoint |
|---|---|---|---|---|---|
| | | | 0–1 years | | |
| c. 2773 A > C, *p.* (R925G) | female | DCM | 0.03 | pathogenic | |
| c. 2773 A > C, *p.* (R925G) | female | DCM | 0.08 | pathogenic | deceased |
| c. 5570 A > C, *p.* (D1857A) | male | DCM | 0.13 | VUS | |
| c. 5635 A > G, *p.* (K1879E) | male | DCM | 0.33 | pathogenic | TX |
| c. 2660 T > G, *p.* (L887R) | female | DCM | 0.67 | pathogenic | |
| c. 1156 T > C, *p.* (Y386H) | male | HCM | 0,01 | likely pathogenic | lvoto |
| c. 3158 G > A, *p.* (R1053Q) | male | HCM | 0.25 | pathogenic | |
| c. 1357 C > T, *p.* (R453C) | female | HCM | 0.33 | pathogenic | ICD |
| c. 5401 > A, *p.* (E1801 K) | male | LVNC | 0,05 | pathogenic | preTX |
| c. 3179 A > G, *p.* (K1060R) | male | carrier | 0.33 | VUS | |
| c. 3179 A > G, *p.* (K1060R) | female | carrier | 0.33 | VUS | |
| c. 4717 G > A, *p.* (E1573K) | male | carrier | 0.75 | VUS | |
| | | | 1–12 years | | |
| c. 2526 T > A, *p.* (S842R) | female | HCM | 5 | pathogenic | TX |
| c. 1816 G > A, *p.* (V606M) | female | HCM | 7 | pathogenic | |
| c. 1357 C > T, *p.* (R453C) | female | HCM | 7 | pathogenic | ICD |
| c. 3158 G > A, *p.* (R1053Q) | male | HCM | 7.5 | pathogenic | |
| c. 1357 C > T *p.* (R453C) | male | HCM | 9 | pathogenic | ICD |
| c. 1987 C > T, *p.* (R663C) | female | HCM | 12 | pathogenic | |
| c. 2155 C > T, *p.* (R719W) | female | HCM | 12 | pathogenic | ICD |
| 1816 G > A, *p.* (V606M) | male | HCM | 12 | pathogenic | |
| c. 1987 C > T, *p.* (R663C) | female | carrier | 2 | pathogenic | |
| c. 1987 C > T, *p.* (R663C) | male | carrier | 4 | pathogenic | |
| c. 3158 G > A, *p.* (R1053Q) | male | carrier | 4.5 | pathogenic | |
| c. 1988 G > A, *p.* (R663H) | male | carrier | 6 | pathogenic | |
| c. 3158 G > A, *p.* (R1053Q) | male | carrier | 12 | pathogenic | |
| c. 2539_2541delAAG, *p.* (K847del) | female | carrier | 12 | likely pathogenic | |

**Table 1.** *Cont.*

| Gene | Sex | CMP | Age at the Time of Dg (Yrs) | Classification | Endpoint |
|---|---|---|---|---|---|
| | | | >12 years | | |
| c. 4285 A > T, *p*. (M1429L) | female | DCM | 57 | VUS | ICD |
| c. 2155 C > T, *p*. (R719W) | female | HCM | 15 | pathogenic | ICD |
| 1816 G > A, *p*. (V606M) | female | HCM | 14.5 | pathogenic | LVOTO + ICD |
| c. 1358 G > A, *p*. (R435H) | male | HCM | 14.5 | pathogenic | ICD |
| c. 1741 C > A, *p*. (H581N) | female | HCM | 15 | VUS | ICD |
| c. 2155 C > T, *p*. (R719W) | female | HCM | 15 | pathogenic | |
| c. 1820 G > A, *p*. (G607D) | male | HCM | 16 | VUS | ICD |
| c. 2155 C > T, *p*. (R719W) | female | HCM | 17 | pathogenic | TX |
| c. 3158 G > A, *p*. (R1053Q) | male | HCM | 18 | pathogenic | |
| c. 2162 G > A, *p*. (R721K) | male | HCM | 20 | VUS | TX |
| c. 1816 G > A, *p*. (V606M) | male | HCM | 33 | pathogenic | deceased |
| c. 3158 G > A, *p*. (R1053Q) | male | HCM | 37 | pathogenic | |
| c. 3179 A > G, *p*. (K1060R) | male | HCM | 38 | VUS | |
| c. 1987 C > T, *p*. (R663C) | male | HCM | 40 | pathogenic | |
| c. 1816 G- > A, *p*. (V606M) | male | HCM | 41 | pathogenic | LVOTO |
| c. 1988 G- > A, *p*. (R663H) | male | HCM | 41 | pathogenic | ICD |
| c. 3158 G > A, *p*. (R1053Q) | male | HCM | 43 | pathogenic | |
| c. 3158 G > A, *p*. (R1053Q) | male | HCM | 43 | pathogenic | |
| c. 3158 G > A, *p*. (R1053Q) | female | HCM | 45 | pathogenic | ICD |
| c. 3158 G > A, *p*. (R1053Q) | female | HCM | 60 | pathogenic | |
| c. 1816 G > A, *p*. (V606M) | male | HCM | 62 | pathogenic | |
| c. 335 G > A, *p*. (W112 *) | female | LVNC | 61 | VUS | |

HCM = hypertrophic cardiomyopathy, DCM = dilated cardiomyopathy, LVNC = left ventricular non-compaction cardiomyopathy, LVOTO = left ventricular outflow tract obstruction, ICD = implantable cardiac defibrillator, preTX = pre-transplantation examination, TX = cardiac transplantation.

The study outcomes and factors associated with more severe disease forms were sudden cardiac death (deceased), cardiac transplantation (TX), pre-transplantation examinations (preTX), LVOTO, or implantable cardiac defibrillator (ICD).

All statistical analyses were performed using SPSS v. 25 (IBM Corporation). Continuous variables are described as mean ± standard deviation or median (range) where appropriate, with three group comparisons conducted using ANOVA or Wilcoxon Rank Sum. Categorical variables were compared using the Chi-Square test.

*Ethics*

The patients older than ten years of age or the parents of younger children gave written informed consent. The Ethics Committee of Helsinki and Uusimaa Region Hospital approved the study plan (ethical license HUS/2227/2018, HUS 291/13/03/03/2008, HUS/3225/2018).

## 3. Results

We identified 48 patients with MYH7 variants; 21 (43.8%) were index patients, 27 (56.3%) were first-degree relatives (nine were not related to any other study patients); 56.3% were men, and 43.8% were women. Table 2 shows the general characteristics of the study population divided by age (0–1, 1–12, and >12 years). Table 1 lists the variant and phenotype of each patient. A family history of the MYH7 variant or SCD was positive at the time of presentation in 27 (56.3%) and 5 (10.4%) patients, respectively. Four patients (8.3%) reported unexplained syncope, and five (10.4%) had neurological deficits at baseline.

All the patients underwent resting and ambulatory ECGs as part of routine CMP protocol. Two patients had atrial fibrillation, six had left ventricular hypertrophy, and four had right bundle branch block. No signs of myocardial infarction or conduction disease, for example, a typical Brugada ECG pattern, were noted.

**Table 2.** Clinical characteristics of patients with MYH7 related cardiomyopathy.

| | Whole Cohort | 0–1 Years | 1–12 Years | >12 Years | *p*-Value |
|---|---|---|---|---|---|
| Age of onset, years | 18.2 (±19.1) | 0.27 (±0.24) | 8.2 (±3.5) | 35.6 (±16.3) | |
| Female sex, *n* (%) | 21 (43.8) | 5 (23.8) | 7 (33.3) | 9 (42.9) | 0.961 |
| FHx CMP, *n* (%) | 27 (56.3) | 8 (29.6) | 12 (44.4) | 7 (25.9) | 0.015 |
| FHx SCD, *n* (%) | 5 (10.9) | 2 (40.0) | 1 (20.0) | 2 (40.0) | 0.707 |
| Unexplained syncope, *n* (%) | 4 (9.8) | 0 (0) | 0 (0) | 4 (100) | 0.044 |
| Neurological deficits, *n* (%) | 5 (10.4) | 1 (20.0) | 3 (60.0) | 1 (20.0) | 0.166 |
| B Blockers (at the time of diagnosing), *n* (%) | 26 (54.2) | 8 (30.8) | 6 (23.1) | 12 (46.2) | 0.647 |
| Echocardiography | | | | | |
| LVMWT, mm | 13.0 (±8.4) | 5.7 (±3.9) | 10.1 (±4.7) | 18.8 (±8.4) | |
| LVEDD, mm | 39.3 (±10.3) | 30.2 (±8.2) | 35.0 (±8.1) | 46.8 (±6.8) | |
| Ejection fraction, % | 64.6 (±12.2) | 50.0 (±20.5) | 72.6 (±4.7) | 66.7 (±12.2) | |
| CMR, *n* (%) | 26 (55.3) | 9 (34.6) | 4 (15.4) | 13 (50.0) | |
| LGE, % [median (range)] | 4.9 (0–24) | 0 (0–24) | 7.5 (0–15) | 10.0 (0–24) | |
| GLS, % [median (range)] | −22.8 [−10–(−33)] | −24.4 [−18–(−30)] | −23.2 [−15–(−27)] | −19.1 [−10–(−33)] | |
| | | | | | |
| Cardiomyopathies, *n* (%) | | | | | |
| HCM | 31 (64.6) | 3 (9.7) | 9 (29.0) | 19 (61.3) | <0.001 |
| DCM | 6 (12.5) | 5 (83.3) | 0 (0) | 1 (16.7) | <0.001 |
| LVNC | 2 (4.2) | 1 (50) | 0 (0) | 1 (50) | 0.613 |
| Carrier | 9 (20.0) | 3 (33.3) | 6 (66.7) | 0 (0) | 0.005 |
| | | | | | |
| Endpoint, *n* (%) | 21 (43.8) | 5 (23.8) | 5 (23.8) | 11 (52.4) | 0.517 |
| Cardiac transplant, *n* (%) | 5 (10.4) | 1 (20.0) | 1 (20.0) | 3 (60.0) | 0.734 |
| Pre-transplant, *n* (%) | 1 (2.1) | 1 (100) | 0 (0) | 0 (0) | 0.216 |
| Death, *n* (%) | 2 (4.2) | 1 (50) | 0 (0) | 1 (50) | 0.551 |
| LVOT obstruction, *n* (%) | 4 (8.3) | 2 (50) | 0 (0) | 2 (50) | 0.287 |
| ICD implantation, *n* (%) | 11 (22.9) | 1 (9.1) | 4 (36.4) | 6 (54.5) | 0.378 |
| Time from diagnose to the endpoint, years | 8.0 (±8.6) | 3.1 (±5.4) | 6.1 (±3.3) | 11.0 (±10.4) | 0.204 |
| Time from birth to endpoint, years | 25.9 (±21.0) | 3.3 (±5.5) | 14.9 (±3.5) | 41.1 (±17.1) | <0.001 |

Data expressed in mean ± standard deviation, unless otherwise indicated. FHx = family history, HCM = hypertrophic cardiomyopathy, DCM = dilated cardiomyopathy, LVNC = left ventricular non-compaction, FHx = family history, LVMWT = left ventricular maximal wall thickness, CI = 95th confidence interval, LVEDD = Left ventricular external end-diastolic diameter, EF = ejection fraction, CMR = cardiac magnetic imaging, LVOT = left ventricular outflow tract, ICD = implantable cardiac defibrillator.

Phenotype: Patient cardiomyopathy type differed significantly according to age. The median age of disease onset was 12.0 years (15.0 years in HCM and four months in DCM). In patients with an age of onset >12 years, HCM was the most frequent diagnosis present in 19/21 (90.4%) patients, DCM and LVNC were both present in 1/21 (4.8%) patients, and there were no variant carriers with a normal phenotype. In patients with an age of onset between 1 and 12 years, 9/15 (60%) showed HCM, no patients showed DCM or LVNC, and 6/15 (40%) were asymptomatic carriers. In patients with an age of onset younger than

1 year, 5/12 (41.6%) showed DCM, 3/12 (25%) showed HCM, 1/12 (8.3%) showed LVNC, and 3/12 (25%) were asymptomatic carriers.

Imaging: CMR was performed for 26 (52.2%) patients. Late gadolinium enhancement (LGE) in the left ventricle was observed in 13/15 (86.7%) of HCM and in 2/4 (50.0%) of DCM. Median CMR global longitudinal strain (GLS) was −19.2% (−33–(−15)) in HCM patients and −20.6% (−24–(−10)) in DCM patients, $p$ = 0.464. In echocardiography, 5/6 (83.3%) of DCM, 2/2 (100%) of LVNC patients, and 1/31 (3.2%) of HCM patients had reduced ejection fraction (EF) (<45%).

Genetics: Following the American College of Medical Genetics and Genomics (ACMG) reclassification, 38 (79.2%) had a pathogenic/likely pathogenic (P/LP) variant, and 10 (20.8%) had a variant of uncertain significance (VUS). MYH7 variant c. 3158 G > A, $p$. (Arg1053Gln), which is classified as pathogenic, was the most common (26.3%), followed by MYH7 c. 1816 G > A, $p$. (Val606Met) (15.8%), and MYH7 c. 1357 C > T, $p$. (Arg453Cys) (7.9%), both also classified as pathogenic. Most of the disease-causing alterations were missense variants. There was one nonsense mutation and two in-frame variants (Table 1). Most of the MYH7 variants were heterozygous.

Only one patient also had another potential variant (DSG2 c. 2137 G > A, $p$. (Glu713Lys)) in our genetic testing, but the clinical phenotype did not support this variant as pathogenic.

The illustration of the MYH7 gene (Figure 1) presents the MYH7 variants in our cohort color-coded by cardiomyopathy type. The distribution pattern throughout the protein is similar to previous reports, with an enrichment in the head area. We highlighted the most relevant functional domains, such as the converter domain and actin-binding regions, to assess the association between the type of CMP and MYH7 variants/domains.

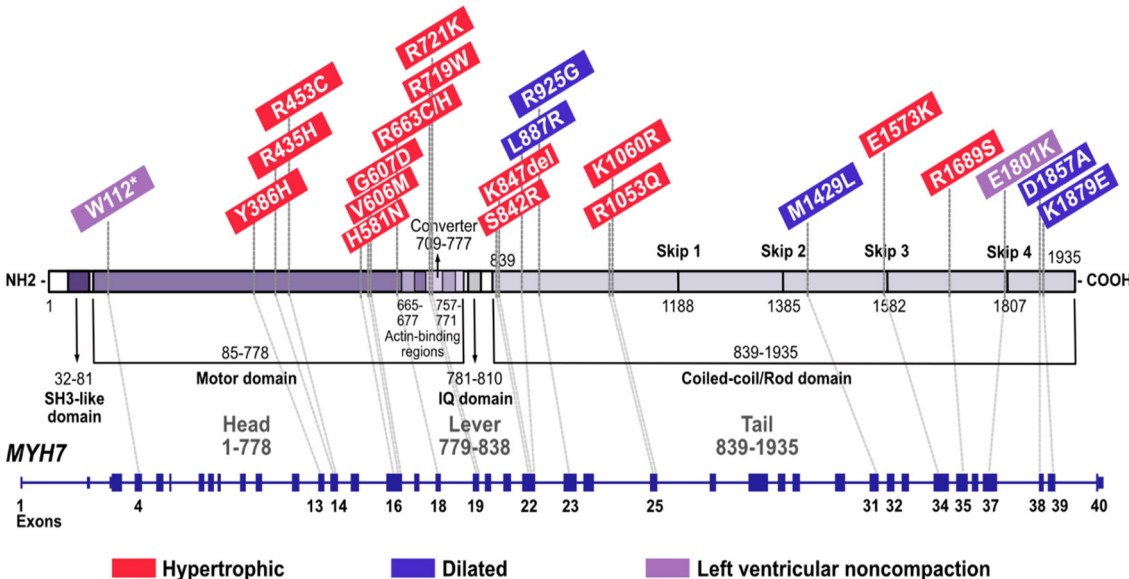

**Figure 1.** Schematic representation of the Myosin-7 protein and the patient mutations analyzed in this study.

The localization of mutations is depicted in relation to important functional domains of the protein. The broad domains of Myosin-7 were described as the head (aa 1–778), lever or neck (aa 779–838), and tail (aa 839–1935), or as subfragment-1 (aa 1–847), subfragment-2 (aa 848–1216), and light meromyosin (aa 1217–1935). They contain functional domains such as SH3-like domain, motor domain, actin-binding regions, converter, IQ domain, and the coiled-coil domain.

Outcome: Factors associated with more severe disease forms were observed in 21/48 (43.8%) of the patients, including 11 patients who underwent ICD implantation (seven as primary and four as secondary prevention), five who underwent cardiac trans-

plantation, two who died, and two who had LVOTO (Figure 2). The mean patient age at the primary outcome event was 26 years and the time from diagnosis to the primary outcome event was 8.0 years, respectively (Table 2). Factors associated with more severe disease forms were observed in 17/38 (45.0%) of P/LP patients and in 4/10 (40.0%) of VUS variants (*p* = 0.788).

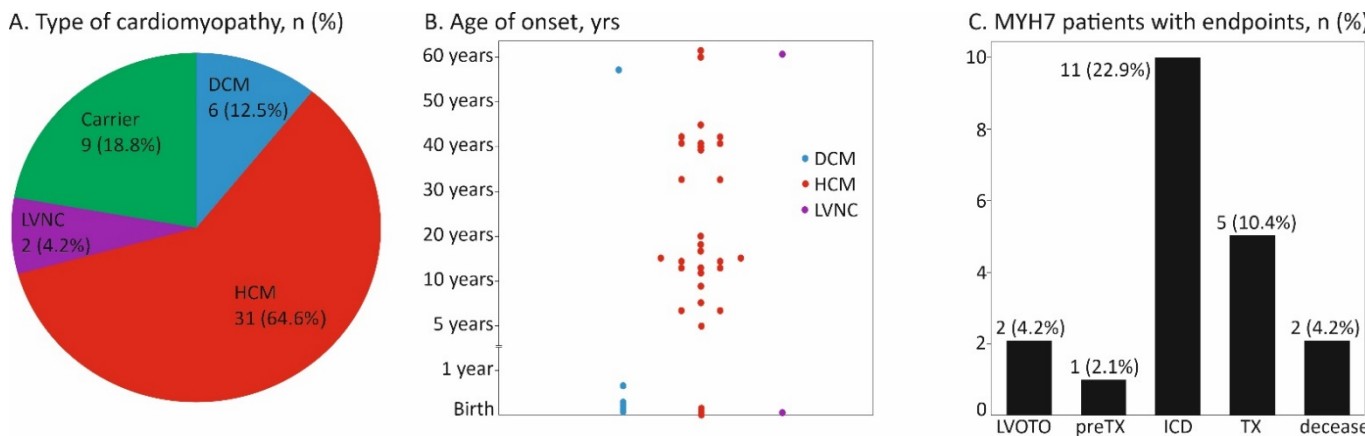

**Figure 2.** (**A**) Patients by cardiomyopathy (CMP) types. (**B**) Age at diagnosis color-coded by CMP type. (**C**) The number of patients with endpoints. PreTX = pre-transplantations examination, lvoto = left ventricular outflow tract obstruction, ICD = implantable cardioverter defibrillator, TX = cardiac transplant.

Pathogenic variants located in the converter domain of MYH7 were associated with particularly adverse outcomes. Amongst our studied patients, two variants were located in the converter domain: c. 2155 C > T, *p*. (Arg719Trp) and c. 2162 G > A, *p*. (Arg721Gln). Of these, one patient underwent a cardiac transplant, and one underwent an ICD implantation.

## 4. Discussion

Our study demonstrates that MYH7 patients under the age of 12 years have a similar symptom burden to older patients. Interestingly, we observed a subgroup of infants with MYH7 associated severe DCM phenotype. We also show clustering of mutations in the MYH7 gene map, highlighting that all pathogenic variants associated with DCM in the present cohort reside in the coiled-coil domain. The number of variants in our cohort is too small to predict domain-specific effects but can contribute to future protein structure-function analyses. For example, the age of onset for five out of six patients with DCM was under one year, and they also presented pathogenic variants in the coiled-coil domain. Future structural–functional analyses could determine why these variants are particularly damaging for MYH7 protein.

Overall, over 800 MYH7 variants were classified as disease-causing, almost all of which are missense variants (HGMD®, available via http://www.hgmd.org (Accessed on 22 Decembar 2021)). The pathogenicity of truncating protein variants is uncertain. We mostly observed missense variants in our cohort, but one variant (MYH7 c. 335 G > A, *p*. (Trp112*)) caused premature truncation. This patient was diagnosed at the age of 61 with LVNC presenting with cardiac symptoms (e.g., palpitations and ventricular premature complexes, shortness of breath), and her LV EF was 40%, although her overall cardiac function was stable. The clinical significance of the MYH7 c. 335 G > A, *p*. (Trp112*) remains unknown. Usually, single truncations are not disease-causing but can associate with very severe disease when they are coincident with a disease-causing missense variant in the second allele.

Our study's most common MYH7 variant was c. 3158 G > A, *p*. (Arg1053Gln), which is the third most common pathogenic variant encountered in Finnish patients with HCM, explaining up to 7.9% of Finnish HCM cases in a recent study [14]. Because of its high

prevalence in Finnish patients with HCM, this pathogenic variant is likely to be a founder variant as in many other genetic diseases in the Finnish population [15]. One patient with the *p*. (Arg1053Gln) variant underwent ICD implantation in our cohort.

In order to further investigate the MYH7 gene and its related mutations, we drew a gene map highlighting the distribution of variants in relation to clinical phenotypes (Figure 1). The MYH7 protein is often divided into three different structural regions: head, neck, tail. The head (aa 1–778) of the protein corresponds in the gene from exon 3 to partway through exon 21, the neck (aa 779–838) extends from exon 21 to 25, and the tail (aa 839–1935) comprises exons 25–40. The functional sites for the ATP-binding domain are encoded between exons 5–12, the actin-binding domain from 13 to 16, the converter region from 18 to 19, and the light chain-binding region from 21 to 22. In our gene map, clustering of the mutations is apparent around some of these sites, although the number of pathogenic variants analyzed is relatively small. A similar finding was also reported in a previous study where it was also noted that variants in the head region were more likely to lead to more severe disease [16–23]. In accordance with this, we noticed that 11 out of 21 primary outcome events (52.4%) were among patients whose variants were in the head region.

The head region contains functional sites for muscle contraction and thus plays a significant role in the functioning of the molecule. It is plausible that the mutation in this region could lead to a detrimental effect on the ability of myosin to fulfill its role in muscle contraction and therefore lead to more severe disease. An especially interesting functional site in the head region is the converter domain located between amino acids 709–777 [23]. This domain is essential for elastic distortion of the cross-bridge. Elastic distortion of a structural element is fundamental to the ability of myosin to generate motile forces. Recently, amino acids between 716 and 719 in this domain were linked to severe forms of CMP [23]. In our study, there were two patients with a variant (c. 2155 C > T, *p*. (Arg719Trp)) between amino acids 716–719. This variant was previously highlighted as entailing a high risk of early mortality. Both patients had HCM, and one of them also underwent an ICD implantation. This mutation affects the converter and light-chain-binding domain, making it more resistant to elastic distortion, thus affecting the ability of myosin to generate motile force [24]. The patient with another variant in the converter domain (c. 2162G > A, *p*. (Arg721Gln)) underwent a cardiac transplant operation at the age of 42. The variant was categorized as VUS.

According to recent findings, the variants related to the DCM-phenotype were clustered in the tail and neck regions [22]. Variants in the tail region may sometimes predispose to a more severe form of cardiomyopathy. This could be due to amino acid substitution that disrupts the assembly or the structure of the thick filament component of the sarcomeres [22]. However, more studies are needed to draw a further conclusion from this finding.

In our cohort, the phenotype of MYH7-related CMP was highly variable. DCM was more prevalent among patients under one year of age, and the prevalence of HCM increased with age. Although DCM is typically an adult-onset disease, the onset may already occur in infancy. Similar findings were also reported previously [25–27]. Our study describes high symptom burden and variable cardiac phenotype on MYH7 patients under 12 years, including severe hypertrophy, malignant ventricular arrhythmias, and cardiac transplantations. It highlights infants who might present a more severe form of CMP and have a poorer prognosis [28–30]. However, it is noteworthy that, in our study, 60% of the patients under one year of age responded well to heart failure medication, and their cardiac function recovered. One patient from this age group underwent cardiac transplantation, and one patient died.

There were no significant differences in any primary outcome events stratified by age (0–1, 1–12, >12 yrs), suggesting that there might already be a considerable symptom burden among patients under 12 years of age with the MYH7 gene mutation. However, more research is needed to verify these results.

The American College of Cardiology (ACC)/American Heart Association (AHA) recently modified the recommendation for family screening of first-degree relatives of affected probands with HCM in the absence of malignant family history. The age threshold for starting family screening [8,31–33] was eliminated. Our findings are in accordance with this recommendation. From the diagnostic point of view, echocardiography remains the cornerstone of cardiac assessment in CMPs. CMR is mainly recommended to describe the phenotype and diagnosis and is valuable for risk stratification for sudden cardiac death [34–36].

Altogether, the prevalence of outcomes was high in our cohort, considering that many variants were classified as VUS (*n* = 10 (20.8%)). Historically, VUS was not considered when evaluating genotype-outcome association, as the pathogenicity of these variants is uncertain. However, a more significant variant burden was found in pediatric HCM patients, including pathogenic and VUS, a burden associated with worse outcomes [36,37]. These data suggest that some of the VUSs with a pathogenic pair are truly disease-causing while many are not. Thus, VUSs should not be used for risk stratification.

## 5. Conclusions

We noticed a subgroup of infants with MYH7 associated severe DCM phenotype. We further demonstrate that patients under the age of 12 years have a similar symptom burden to older patients. These findings support the observations that clinical and genetic screening should also be considered for younger MYH7 family members to identify patients needing closer monitoring and interventions. However, more research is required to estimate causal relationships, considering our limited study sample.

**Author Contributions:** T.V.: data acquisition, processing, and interpretation; writing of the manuscript. T.O., T.H.: study concept and supervision, data interpretation, critical revision of the manuscript. L.M.: Data interpretation, critical revision of the manuscript. S.W.: data acquisition, critical revision of the manuscript. J.K.: genetic analysis, critical revision of the manuscript. A.H.: critical revision of the manuscript. C.V.: genetic analysis, data interpretation, critical revision of the manuscript. All authors have read and agreed to the published version of the manuscript.

**Funding:** This work was financially supported by the State funding for university-level health research (TYH2020210, Y2020SK004, Y1016SK004), the Foundation for Pediatric Research Center, Aarne Koskelo Foundation, and the Finnish Foundation for Cardiovascular Research.

**Institutional Review Board Statement:** Written informed consent was obtained from the patients (ethical license HUS/2227/2018, HUS 291/13/03/03/2008, HUS/3225/2018).

**Informed Consent Statement:** Informed consent was obtained from all subjects involved in the study.

**Data Availability Statement:** The data that support the findings of this study are available from the corresponding authors upon reasonable request.

**Acknowledgments:** We thank Anu Suomalainen for their valuable help in the interpretation of study data. We thank patients for participation in the study.

**Conflicts of Interest:** Juha Koskenvuo is a full-time Executive Director, Lab and Medical, at Blueprint Genetics. Other authors declare no conflict of interest.

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
