# Peer review of "MYH7 Genotype–Phenotype Correlation in a Cohort of Finnish Patients"

_cardiogenetics, doi:10.3390/cardiogenetics12010013_

Round 1

Reviewer 1 Report

The study aimed to analyze the clinical presentation and natural history of patients with MYH7-related cardiomyopathy with special reference to patients under 12 years. The authors studied a cohort of patients who underwent cardiac transplantations and selected those carrying a MYH7 variant, highlighting the differences between patients aged <12 years and those older.

The authors found that patients under 24 the age of 12 years have a similar symptom burden compared to older patients. However, they noticed that infants with MYH7 23 are a significant DCM phenotype-associated subgroup.

I do not know the group of investigators, the paper is interesting however English needs some revision.

I think that the paper contains interesting genotype-phenotype correlations: they are very useful for biologists and clinicians that want to better characterized the variants.

However, I have some concerns about the methodology. This is a very selected population: the patients were all referred for cardiac transplantation therefore the phenotype is more severe than the CMPs general population. Moreover, the genetic testing was performed with Sanger sequencing on selected genes based on phenotype. I am wondering which genes have been exactly tested according to the phenotype and I am wondering whether other variants on other genes not screened may have played a role on the phenotype. Please, may you specify better the genetic analysis (methods and results)? I find it strange that no patients had other variants on other genes.

OTHER COMMENTS

In general, you should clarify better the population you studied. Also in the title. I would suggest changing it in MYH7 genotype-phenotype correlation in a cohort of Finnish patients who underwent cardiac transplantation.

In the Methods, you should specify which protocol was performed for the imaging investigations, who performed those investigations and which machines were used (both for echo and MRI)

Also, did the patients undergo 24-hour Holter ECG monitoring? Was an ECG performed? Which were your findings? If not, why?

MINOR COMMENTS

METHODS:

Page 2, line 57: “region beginning from 2002” when did the inclusion end?

Page 2, lines 83-85. What was the definition of SCD? (see comment on results)

RESULTS:

Page 3, line 110: “We identified 43 patients with MYH7 variants” How many patients did you screen in total?

Page 3, lines 113-114 “A family history of MYH7 variant or SCD was positive at 113 the time of presentation in 25 (58.1%) and 4 (9.5%) of patients respectively”. Please , specify in the methods your criteria for family history of SCD

Page 5 lines 136-137: “. The median late gadolinium 136 enhancement (LGE) percentage of the LV wall was 6.0 % in the whole study population” what does it mean? Is it the percentage of LGE related to myocardial mass or the percentage of patients in which you detected LGE? If you have quantified the LGE, you should specify in the methods how did you quantify the LGE. Considering the limited value of the LGE quantification (especially in diseases other than HCM) I would rather prefer to know of many HCM/DCM/LVNC had LGE.

Author Response

We want to thank the reviewer for taking the time to assess our manuscript. We have addressed all the concerns that the reviewer raised, and we feel that the manuscript is greatly improved as a result. Please find attached our response to reviewer. 

Reviewer 2 Report

I have been asked to review this manuscript on the prevalence and outcomes of Finnish MYH7 patients.

This is an interesting manuscript that has studied Finnish MYH7 mutations in cardiomyopathy patients over almost two decades.

There are however some points that need clarification.

The first is the use of term prevalence in the title. The authors have not mentioned any denominator that is the total number of cardiomyopathy patients studied for mutations in MYH7 thus the prevalence cannot be calculated. Genotyping was carried out discretion of the treating clinician. Thus, the term prevalence should be deleted from the title. A more appropriate title should be used.

Similarly, the abstract should be clear on the methodology used. The authors state that ‘We analyzed the clinical presentation and natural history of patients with MYH7-related cardiomyopathy with special reference to patients under 12 years.’ However, there are 3 subgroups- 0-1 years, 1-12 years and >12 years. This should be stated in the abstract.

The lines in the abstract ‘Our results characterize a clinically significant MYH7 cohort, emphasizing 22 the high variability of the CMP phenotype depending on age. We noticed that infants with MYH7 23 are a significant DCM phenotype-associated subgroup.’ are not clear. The major results and conclusion should be clear from the abstract.

Materials and methods: This is not a cross sectional study as mentioned by authors. Cross sectional studies look at data at one point of time. This study included patients over several years.

The methods should be clear. The authors state that the study included patients from 2002. What was the endpoint of time.

Primary endpoints included LVOTO and implantable cardiac defibrillator. These should not be primary outcomes. Implantation of ICD is not a primary outcome but an appropriate ICD discharge can be.

Results: the authors state that ‘We identified 43 patients with MYH7 variants; 18 (41.9%) were index patients’. Were the rest family members. This should be clearly stated.

Conclusions are not clear. ‘Our results characterize a clinically significant MYH7 cohort, emphasizing the high variability of the CMP phenotype depending on age’. This statement should be clarified. What is the relationship morphology with age?

‘Our study suggests that early clinical and genetic screening should be considered for younger MYH7 family members to identify patients needing closer monitoring and interventions.’ This conclusion is a general statement not based on this studies result.

Table 1: Terms dg and yrs should have the full form mentioned.

Table 2: One endpoint is preTX. What is this. This is not mentioned in methods. Another unclead endpoint is exitus.

Overall, the language needs to be improved.

Author Response

(The authors gave the same response as above.)

Reviewer 3 Report

The present study sought to evaluate the high variability of the CMP phenotype in patients with MYH7 mutation, with special reference to patients under 12 years. Precision medicine aims to achieve improved survival by strategies that recognize the genetic and phenotypic individuality of patients and stratify treatment accordingly and genetic cardiomyopathies represent an ideal disease group to fully embark on this concept . At this regard, this study is very interesting because it highlights how the localization of mutation which is associated with important functional domains of protein probably correlates with clinical phenotypes and severity of disease.

Major and minor comments:

  • The study also highlights how MYH7 mutation in a subgroup of patients under 1 year is prevalently associated with DCM. Nevertheless, you have considered a small number of patients in order to establish a real correlation rather than casual evidence and it can not representative of real world. Your study population is too small to claim the association between phenotype and age of cardiomyopathy onset.
  • Cardiac magnetic resonance (CMR) allows a morphological evaluation of the associated (and sometimes pathognomonic) cardiac findings of any form of cardiomyopathy (Cardiac magnetic resonance in hypertrophic and dilated cardiomyopathies, Silvia Pradella et al., Cardiac radiology). We know that is a cross sectional retrospective analysis so patients may had been managed differently but I would specify how many patients have performed MRI in addition to echocardiogram and the missing data about MRI in the other group of patients evaluated only by echocardiogram.
  • It would be interesting to investigate if there is any cardiac conduction disease associated with the cardiomyopathy. This paper should be quoted and commented: Mutations in MYBPC3 and MYH7 in Association with Brugada Type 1 ECG Pattern: Overlap between Brugada Syndrome and Hypertrophic Cardiomyopathy? Marianna Farnè, Cristina Balla, Cardiogenetics). There are some evidences showing that mutation in MYH7 gene are related to Brugada syndrome as well as to hypertrophic cardiomyopathy. So, it would have been better, in our opinion, to evaluate the presence of conduction disease by performing EKG in every patient under investigation in your study.
  • Page 21 lines 16-18 in Materials and Methods: “The following MRI criterion was used; the end-diastolic ratio of non-compacted to compacted layers above 2.3”. You should specify that this parameter is measured in short axis and we refer to it as Regional LVNC, in contrast to Global LVNC which is defined as “the involvement of greater than 7 NC segments of the 17 segment AHA model” (Improving the diagnosis of LV non compaction with cardiac magnetic resonance imaging, P. Choudhary et al., International Journal of Cardiology)
  • About LV outflow tract pressure gradient in LVOTO, you should specify that it is at rest or during physiological provocation such as Valsalva manoeuvre.
  • Variant assessment has become the bottleneck of large scale sequencing tests. Final interpretation of the clinical significance of a variant requires examination of all the available evidence. While some data can be strong enough to determine or rule out pathogenicity, most information only moderately influences final conclusions and is valuable in combination (Large next-generation sequencing gene panels in genetic heart disease: yield of pathogenic variants and variants of unknown significance, F.H.M. van Lint et al., Neth Heart J). As you have underlined, the pathogenicity of VUS is uncertain, so the inclusion of these variants in such a small population is rather questionable.
  • In this study 9 patients underwent ICD implantation. I would suggest to specify which patients were implanted in primary and secondary prevention respectively.
  • It could be interesting to report some details about pre transplantation examination and the final destination of those patients
  • Why have you considered a cut off of 12 years to divide your study population?
  • In conclusion, you claim that there is a high variability of the CMP phenotype depending on age; established that genetic alterations in the tail are related to DCM phenotype, whilst the genetic alterations in the head are related to HCM phenotype, is more likely that different phenotypes depend on the site of genetic mutation rather than age. So it would be interesting to continue to investigate in this sense with future studies.

Author Response

(The authors gave the same response as above.)

Round 2

Reviewer 1 Report

Thank you.